# Microwave-Assisted Hydrothermal Synthesis of Zinc-Aluminum Spinel ZnAl_2_O_4_

**DOI:** 10.3390/ma15010245

**Published:** 2021-12-29

**Authors:** Tomasz Strachowski, Ewa Grzanka, Jan Mizeracki, Adrian Chlanda, Magdalena Baran, Marcin Małek, Marlena Niedziałek

**Affiliations:** 1Research Group of Graphene and Composites, Lukasiewicz Research Network–Institute of Microelectronics and Photonics IMiF, Al. Lotnikow 32/46, 02-668 Warsaw, Poland; adrian.chlanda@imif.lukasiewicz.gov.pl (A.C.); magdalena.baran@imif.lukasiewicz.gov.pl (M.B.); 2Institute of High Pressure Physics PAS “Unipress”, Sokolowska 29/37, 01-142 Warsaw, Poland; elesk@unipress.waw.pl (E.G.); janekm@unipress.waw.pl (J.M.); 3Faculty of Civil Engineering and Geology, Military University of Technology, ul. Gen. Sylwestra Kaliskiego 2, 00-908 Warsaw, Poland; marcin.malek@wat.edu.pl (M.M.); marlena.niedzialek@wat.edu.pl (M.N.)

**Keywords:** zinc-aluminum spinel ZnAl_2_O_4_, hydrothermal synthesis, microwave reactor

## Abstract

The drawback of the hydrothermal technique is driven by the fact that it is a time-consuming operation, which greatly impedes its commercial application. To overcome this issue, conventional hydrothermal synthesis can be improved by the implementation of microwaves, which should result in enhanced process kinetics and, at the same time, pure-phase and homogeneous products. In this study, nanometric zinc aluminate (ZnAl_2_O_4_) with a spinel structure was obtained by a hydrothermal method using microwave reactor. The average ZnAl_2_O_4_ crystallite grain size was calculated from the broadening of XRD lines. In addition, BET analysis was performed to further characterize the as-synthesized particles. The synthesized materials were also subjected to microscopic SEM and TEM observations. Based on the obtained results, we concluded that the grain sizes were in the range of 6–8 nm. The surface areas measured for the samples from the microwave reactor were 215 and 278 m^2^ g^−1^.

## 1. Introduction

Spinels are a group of compounds with the general formula AB_2_O_4_, where A is a divalent metal (Zn, Mg, Fe or Mn) and B is a trivalent metal (Al, Fe, Cr or Mn). The spinel network cell is regular. It can be described as consisting of a series of F-type networks (embedded corners and wall centers). A regular elementary cell contains 32 oxygen anions and 24 cations: 8 (A)-type cations occupy tetrahedral gaps, and 16 (B)-type cations are in octahedral gaps. In regular spinels (which include ZnAl_2_O_4_), the tetrahedral positions are occupied by 8 A(+III) ions, and the octahedral positions are occupied by 16 B(+III) ions [1,2,3,4,5].

Zinc-aluminum spinel can be obtained by various methods, such as sol–gel [6,7,8,9,10,11,12], hydrothermal methods [13,14,15,16,17,18,19,20], co-precipitation [21,22,23,24,25,26], combustion methods [27,28,29,30] and solvothermal synthesis [31,32,33]. Implementation of these methods resulted in splines with a size distribution ranging from 15 to 100 nm. It is worth noting that hydrothermal synthesis is the simplest method to obtain spinels with nanometric grain size. However, this method usually needs a long process time and low temperature. Conventional hydrothermal synthesis can be improved by the implementation of microwaves, which results in a faster increase in temperature and process kinetics and the generation of pure-phase and homogeneous products.

Oxide spinels comprise a broad group of compounds with complex structures that are of great technological importance. This group includes zinc-aluminum spinel, which is characterized by many desirable properties, including: high mechanical resistivity, high thermal stability and low sintering temperature. Zinc-aluminum spinel is widely used in chemical and electronics industries and in catalytic reactions such as cracking, dehydration, hydrogenation and dehydrogenation. This applies mainly to spinel doped with Fe(+III) ions [34,35,36,37,38,39,40]. The zinc-aluminum spinel is transparent to light at wavelengths above 320 nm and has therefore found application in optoelectronics (energy gap 3.8 eV) [41,42,43,44,45,46,47,48,49,50]. In addition, transparent spinels without defects are used in precision mechanics (e.g., for bearing production) and as gemstones [51,52,53,54,55]. This spinel has also found applications in the manufacture of nanotubes, nanowires and thin films. Zinc-aluminum spinel doped with ions of various chemical elements, such as Co^3+^, Er^3+^, Eu^3+^, Yb^3+^, Tb^3+^ and Mn^3+^, also has interesting properties. Spinel doped with these elements exhibits excellent luminescent properties [56,57,58,59,60,61,62,63,64,65].

This manuscript contains a detailed description of the synthesis and characterization of selected vital properties of as-obtained materials. Although one can find literature reports describing scientific efforts aimed at fabricating similar materials, it is worth underlining that it is hard to obtain zinc-aluminum spinel grains characterized by small sizes (in a range of single nanometers). There are several literature reports of obtaining zinc-aluminum spinel with sizes measured in single nanometers [7,19]. 

In addition to zinc-aluminum spinel, other compounds in the spinel group, such as ZnGa_2_O_4_ [66], MgCr_2_O_4_ [67] and SrAl_2_O_4_ [68], are of great interest. 

The authors consider the most interesting achievement of this research to be the preparation of zinc-aluminum spinel with a small grain size and large specific surface area. In the literature, one can find zinc-aluminum spinels with much larger grains [6,11,12,13,18].

## 2. Materials and Methods

### 2.1. Materials

Zn(NO_3_)_2_*6H_2_O and Al(NO_3_)_3_*9H_2_O were used for the hydrothermal synthesis of zinc-aluminum spinel. A 2 M aqueous solution of KOH was used as the substance for the precipitation of zinc and aluminum hydroxides. When preparing the solution of the respective salts, the proportion was maintained so that the molar ratio of Al:Zn was 2:1.

Appropriate amounts of zinc and aluminum salts were weighed in a 500 mL beaker, and then 150 mL of distilled water was added. Mixing was carried out with a magnetic stirrer at room temperature for 1 h. A 2 M KOH aqueous solution was gradually poured into the mixed solution until pH = 12 was reached. Figure 1 shows a schematic step-by-step description of the production of zinc-aluminum spinel.

The obtained zinc-aluminum hydroxide suspension was poured into reaction vessels and placed in a microwave reactor. The microwave reactor is a laboratory instrument designed for hydrothermal syntheses in a microwave field. Microwave energy is drawn into the head from the magnetron through a waveguide in which a short antenna is immersed. The whole system is controlled by means of a central processing unit (CPU) controller communicating with a personal computer (PC). The computer program allows adjusting power thresholds and pressure limits, along with temperature and pressure registration during the process [69].

The experiment in the microwave reactor was as follows. The prepared solution was poured into a Teflon reaction vessel with a capacity of 110 mL. The vessel was then closed with a Teflon lid and placed inside the stainless-steel head. The final step was to close the head and start the process.

The appropriate process parameters must be set in order to obtain good-quality resulting material. In the case of the presented study, the process was conducted for 15, 30 or 60 min. After completion of the processes, the product was filtered under pressure through filters and washed repeatedly with distilled water. The purified material was then placed in a vacuum dryer (70 °C/24 h).

### 2.2. Methods

A Siemens D-5000 X-ray diffractometer (Siemens-Bruker Corporation, Germany) was used to analyze the obtained synthesis products. It enables the determination of the phase composition, recognition of crystallographic lattice and identification of the coordinates of atoms in the elementary cell (by the Rietveld method).

Density was determined by a gas pycnometer (AccuPyc 1330 helium pycnometer, Micromeritics, Norcross, GA, USA). The specific surface area was measured using the BET adsorption method with a Micromeritics device. Microstructure studies of nanopowders were performed using scanning microscopy with a Zeiss LEO1530 (Carl Zeiss, Germany) equipped with a Zeiss Gemini column.

## 3. Results and Discussion

### X-ray Tomography of Ceramic Preforms

Table 1 presents the results of tests on zinc-aluminum spinel obtained in the microwave reactor. The overriding conclusion from these results is that the grain size was in the same range (6–7 nm), and it was independent of the synthesis time. The process temperature was approximately 200 °C. The density and specific surface areas were maintained at similar levels.

XRD analysis revealed that the microwave reactor yielded a product that contained a ZnO phase in addition to the ZnAl_2_O_4_ spinel phase. In order to obtain a pure spinel phase, we decided to perform an experiment using the same process time while increasing the process pressure (50, 60 and 70 atm). Figure 2 shows the results of XRD analysis for samples obtained at different pressures for 30 min. It can be observed that as the pressure increased, the ZnO phase disappeared, and the pure ZnAl_2_O_4_ phase remained. 

Table 2 shows the results of the analysis of zinc-aluminum spinel samples that were synthesized at elevated pressure. It was observed that as the pressure and time process increased, the ZnO phase disappeared, and the pure ZnAl_2_O_4_ phase remained.

Figure 3 shows the dependence of the specific surface area (BET) as a function of process time and pressure. It can be observed that the highest value was achieved for the sample obtained in the 30 min process at a pressure of 50 atm. As the pressure and process time increased, the value of the specific surface area decreased. This is particularly evident for samples obtained at 60 min of process time. This was significantly influenced by lowering the process temperature, increasing the pressure and using microwaves, which also shortened the process time. Based on the literature review [6,7,8,12,13,14,15,20,21], it can be noted that previous studies carried out the synthesis of ZnAl_2_O_4_ in the temperature range from 600 to 900 °C. At those temperatures, a product with a small specific surface area and, consequently, a large grain size was obtained.

Figure 4 shows the dependence of density on pressure and process time. It can be observed that with an increase in these parameters (up to 30 min), the density increased, while after this time, it decreased.

After measuring the density and specific surface area with the BET method, it was found that 30 min was the optimal process time, as both parameters (density and specific surface area) were characterized by the highest values.

Figure 5 shows scanning microscope (SEM) and transmission microscope photographs (Figure 6) for samples obtained at the optimum process time of 30 min. One can notice the powder crystallites in the form of spherical grains. Observation by transmission microscopy confirmed the actual nanometric grain size obtained in this study.

Based on the obtained results, it can be concluded that in a single synthesis, zinc-aluminum spinel was obtained with a grain size in the range of 6–8 nm and a density in the range of 3.41–3.51 g/cm^3^.

## 4. Conclusions

Hydrothermal synthesis of zinc-aluminum spinel using a microwave reactor was carried out.

We concluded that the syntheses performedin the microwave reactor resulted in zinc-aluminum spinel with an average grain size in the range of 6–7 nm. At a lower pressure (39 atm), a ZnO phase appeared in addition to the spinel phase, which in this case, was unwanted. In order to obtain a pure spinel phase (without ZnO doping), the process pressure was increased (50, 60 and 70 atm).

The obtained products were characterized by a high specific surface area in the range of 220–280 m^2^/g. The density measured on a helium pycnometer was in the range of 3.4–3.7 g/cm^3^. The aforementioned material properties were plotted against time, which enabled the designation of the optimum process time: 30 min.

Analysis of the morphology using a scanning microscope showed that the reaction product agglomerated as beads (Figure 5). These structures were composed of fine crystallites that were attracted to each other in the reaction slurry or during post-synthesis washing. The morphology of the powders observed with a transmission microscope (TEM) enclosed fine grains (below 10 nm). We concluded that the higher the process pressure, the smaller the grains. The finest grains were observed for powder obtained at a pressure of 70 atm and a time of 30 min.

Having all of this in mind, we want to underline that the synthesis of zinc-aluminum spinel using a hydrothermal method using a microwave reactor is advantageous because it results in a final product with:High density;Phase homogeneity;Nanometric grain size.

## Figures and Tables

**Figure 1 materials-15-00245-f001:**
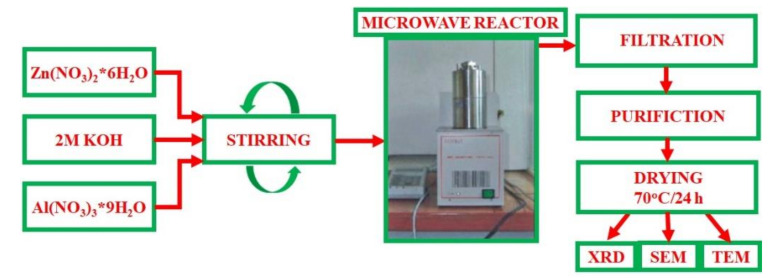
Scheme of the production of zinc-aluminum spinel.

**Figure 2 materials-15-00245-f002:**
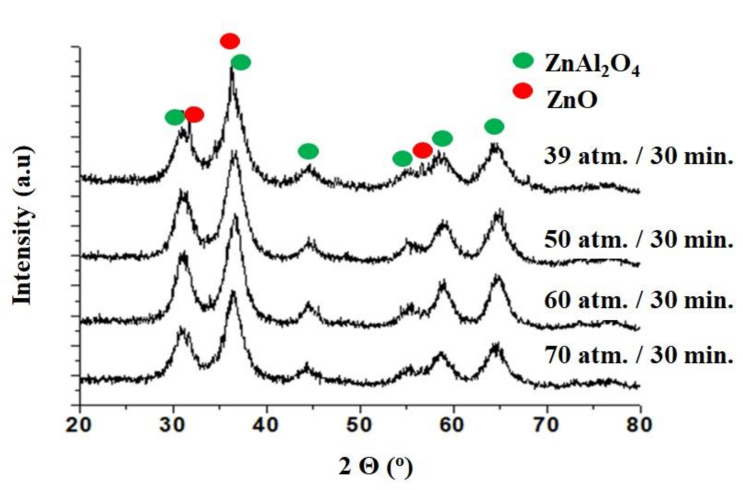
XRD analysis of zinc-aluminum spinel under different pressures.

**Figure 3 materials-15-00245-f003:**
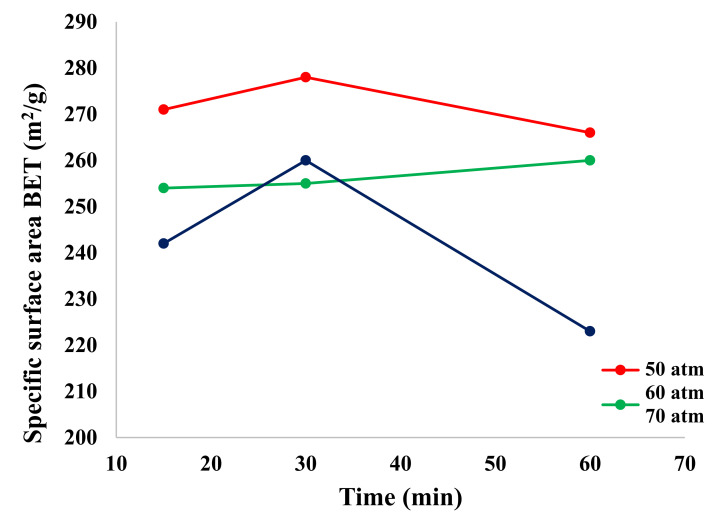
Dependence of the specific surface area as a function of process time and pressure.

**Figure 4 materials-15-00245-f004:**
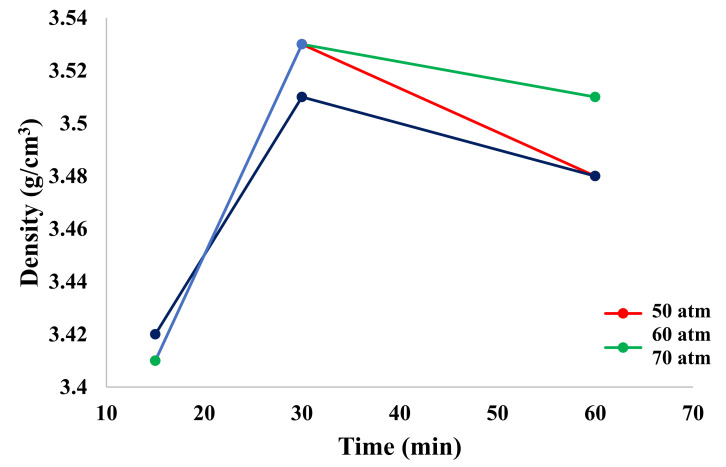
Dependence of density on pressure and process time.

**Figure 5 materials-15-00245-f005:**
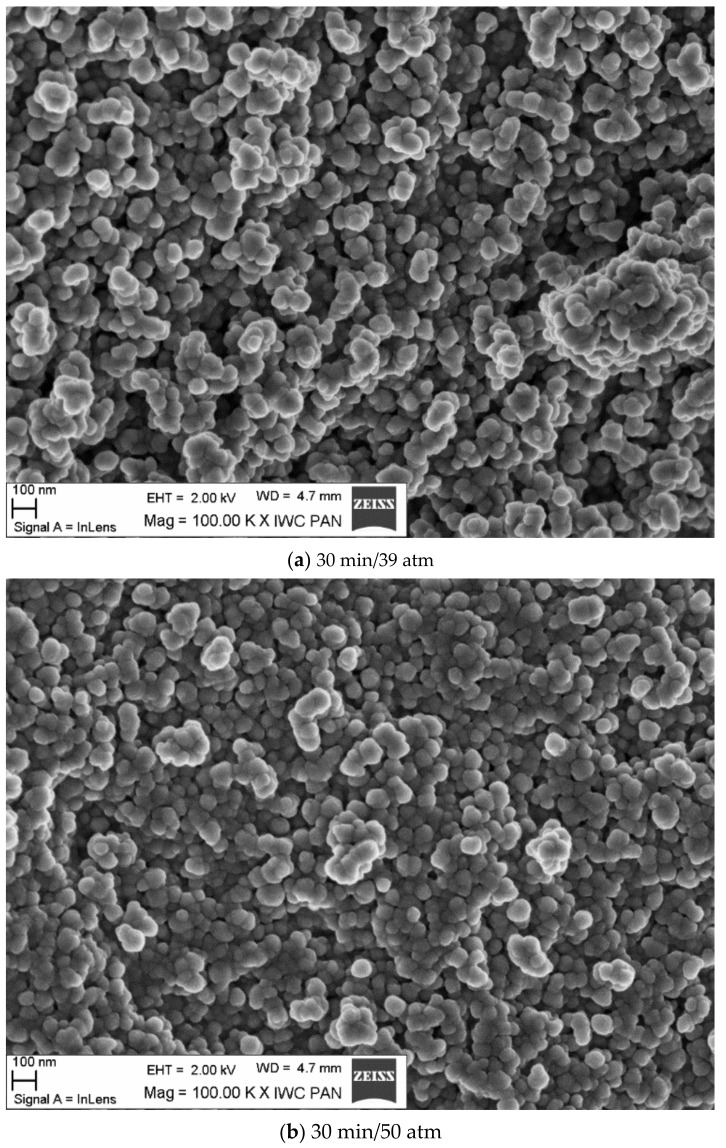
SEM pictures of samples obtained at the optimum process time of 30 min.

**Figure 6 materials-15-00245-f006:**
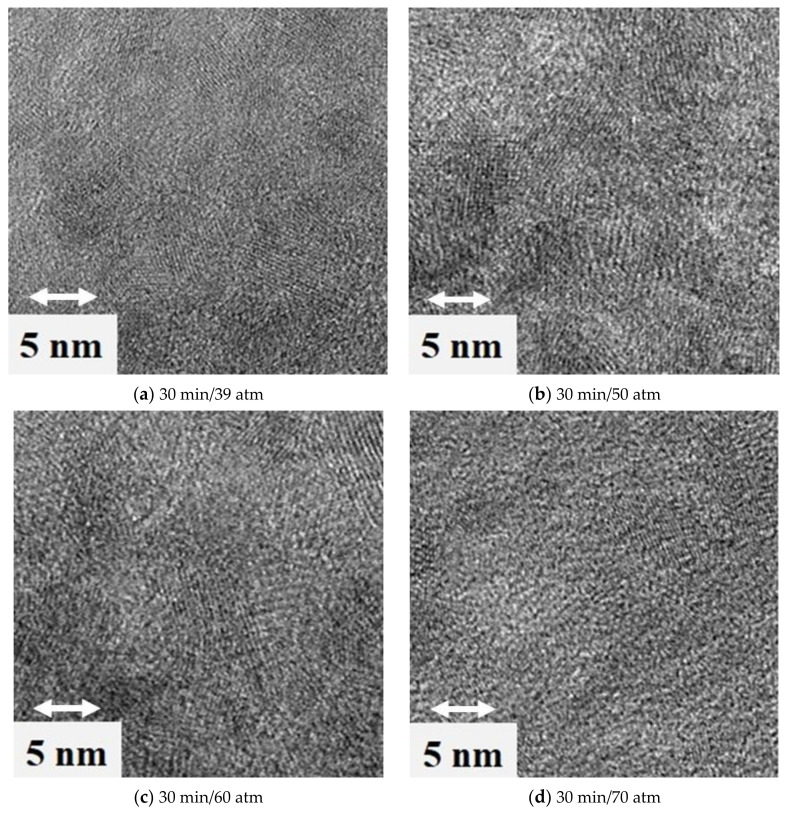
TEM pictures of samples obtained at the optimum process time of 30 min.

**Table 1 materials-15-00245-t001:** Results of zinc-aluminum spinel analysis.

**Time (min)**	15	30	60
**Pressure (atm)**	39	39	39
**Density (g/cm^3^)**	3.72	3.53	3.68
**BET (m^2^/g)**	260	266	262
**Grain size (nm)**	6–7	6–7	6–7
**Phases**	ZnAl_2_O_4_ + ZnO	ZnAl_2_O_4_ + ZnO	ZnAl_2_O_4_ + ZnO

**Table 2 materials-15-00245-t002:** Results of the analysis of zinc-aluminum spinel samples after synthesis.

Sample	Density (g/cm^3^)	BET (m^2^/g)	Grain Size (nm)	Phases
15 min/50 atm	3.41	271	7	ZnAl_2_O_4_
15 min/60 atm	3.41	254	7	ZnAl_2_O_4_
15 min/70 atm	3.42	242	7	ZnAl_2_O_4_
30 min/50 atm	3.53	278	6	ZnAl_2_O_4_
30 min/60 atm	3.53	255	8	ZnAl_2_O_4_
30 min/70 atm	3.51	260	6	ZnAl_2_O_4_
60 min/50 atm	3.48	266	6	ZnAl_2_O_4_
60 min/60 atm	3.51	260	6	ZnAl_2_O_4_
60 min/70 atm	3.48	223	7	ZnAl_2_O_4_

## Data Availability

All individuals included in this section have agreed to be confirmed.

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
