# Peer review of "Microwave-Assisted Hydrothermal Synthesis of Zinc-Aluminum Spinel ZnAl2O4"

_materials, 2021, doi:10.3390/ma15010245_

Round 1

Reviewer 1 Report

In the introduction, the authors should highlight the difficulties in the synthesis of nanoscale spinel and the specific application prospect.

The authors claim ‘although one can find a literature reports describing scientific effords aimed to fabricate similar materials, it is worth to underline that it is hard to obtain zinc-aluminum spinels grains characterized with small size (in a range of single nanometers)’, but do not explain why, and there is no comparison between the results of this work and other literature.

Microwave reactor is a very common experimental equipment. There is no need to introduce so many details.

The quality of article drawings needs to be improved.

The results of material characterization should be discussed in depth.

The language used in scientific papers should be accurate and concise. The author should improve their writing.

Unfortunately, this manuscript has not yet reached the standard of scientific paper.

Author Response

REVIEWER 1

Comments and Suggestions for Authors

In the introduction, the authors should highlight the difficulties in the synthesis of nanoscale spinel and the specific application prospect.

The authors claim ‘although one can find a literature reports describing scientific effords aimed to fabricate similar materials, it is worth to underline that it is hard to obtain zinc-aluminum spinels grains characterized with small size (in a range of single nanometers)’, but do not explain why, and there is no comparison between the results of this work and other literature.

The difficulty related with obtaining small grains of aforementioned materials can be driven with the type of synthesis in which the zinc-aluminum spinel was obtained. Moreover size of the resulting material may also be affected by the treatment of the material after synthesis, such as calcination.

Microwave reactor is a very common experimental equipment. There is no need to introduce so many details.

The Reviewer is right – it is a common laboratory equipment, however there are many types of microwave reactors for synthesizing nano particles. They differ in construction, power, volume of the reaction vessel and even its geometry. These parameters can significantly influence the size of the material obtained and for that reason we decided to go with detailed description of the reactor and parameters implemented in this study.

The quality of article drawings needs to be improved.

Appropriate changes were made in the manuscript.

The results of material characterization should be discussed in depth.

Appropriate changes were made in the manuscript.

The language used in scientific papers should be accurate and concise. The author should improve their writing.

Appropriate changes were made in the manuscript.

Reviewer 2 Report

Referee report on manuscript “Microwave-assisted hydrothermal synthesis of zinc-aluminum spinel ZnAl2O4” by  Tomasz Strachowski et al

This work is devoted to the description of the hydrothermal synthesis of zinc aluminate (ZnAl2O4) with a spinel structure, which is currently one of the promising materials for various luminescent detectors. In this regard, the usefulness of this work is beyond doubt and the article can be recommended for publication after clarifying several points that will undoubtedly only improve this manuscript.

  1. Line 57. Somewhere here it would be absolutely useful to point out that, along with ZnAl2O4, several other similar compounds are under close scrutiny of researchers. These are ZnGa2O4, MgCr2O4, SrAl2O4  See some recent papers for your attention:

Luchechko, A.; Zhydachevskyy, Y.; Ubizskii, S.; Kravets, O.; Popov, A.I.; Rogulis, U.; Elsts, E.; Bulur, E.; Suchocki, A. Afterglow, TL and OSL Properties of Mn2+-doped ZnGa2O4 Phosphor. Sci. Rep. 20199, 9544

Mykhailovych, V.; Kanak, A.; Cojocaru, Ş.; Chitoiu-Arsene, E.-D.; Palamaru, M.N.; Iordan, A.-R.; Korovyanko, O.; Diaconu, A.; Ciobanu, V.G.; Caruntu, G.; Lushchak, O.; Fochuk, P.; Khalavka, Y.; Rotaru, A. Structural, Optical, and Catalytic Properties of MgCr2O4 Spinel-Type Nanostructures Synthesized by Sol–Gel Auto-Combustion Method. Catalysts 202111, 1476.

Zhai, B.-G.; Huang, Y.-M. Green Afterglow of Undoped SrAl2O4Nanomaterials 202111, 2331. https://doi.org/10.3390/nano11092331

  1. Lines 37-39. It would be useful to indicate what particle sizes and with what size distribution these methods give.
  2. Line 115. So exactly 6 nm?  And is there really no size distribution?
  3. How stable were the SEM and TEM pictures with time and were there any features indicating aging of the obtained materials?

Author Response

REVIEWER 2

Comments and Suggestions for Authors

Referee report on manuscript “Microwave-assisted hydrothermal synthesis of zinc-aluminum spinel ZnAl2O4” by  Tomasz Strachowski et al

This work is devoted to the description of the hydrothermal synthesis of zinc aluminate (ZnAl2O4) with a spinel structure, which is currently one of the promising materials for various luminescent detectors. In this regard, the usefulness of this work is beyond doubt and the article can be recommended for publication after clarifying several points that will undoubtedly only improve this manuscript.

  1. Line 57. Somewhere here it would be absolutely useful to point out that, along with ZnAl2O4, several other similar compounds are under close scrutiny of researchers. These are ZnGa2O4, MgCr2O4, SrAl2O4  See some recent papers for your attention:

Luchechko, A.; Zhydachevskyy, Y.; Ubizskii, S.; Kravets, O.; Popov, A.I.; Rogulis, U.; Elsts, E.; Bulur, E.; Suchocki, A. Afterglow, TL and OSL Properties of Mn2+-doped ZnGa2O4 Phosphor. Sci. Rep. 20199, 9544

Mykhailovych, V.; Kanak, A.; Cojocaru, Ş.; Chitoiu-Arsene, E.-D.; Palamaru, M.N.; Iordan, A.-R.; Korovyanko, O.; Diaconu, A.; Ciobanu, V.G.; Caruntu, G.; Lushchak, O.; Fochuk, P.; Khalavka, Y.; Rotaru, A. Structural, Optical, and Catalytic Properties of MgCr2O4 Spinel-Type Nanostructures Synthesized by Sol–Gel Auto-Combustion Method. Catalysts 202111, 1476.

Zhai, B.-G.; Huang, Y.-M. Green Afterglow of Undoped SrAl2O4Nanomaterials 202111, 2331. https://doi.org/10.3390/nano11092331

Thank you very much for your submitted publications and valuable comments. I was happy to include these publications in our manuscript.

  1. Lines 37-39. It would be useful to indicate what particle sizes and with what size distribution these methods give.

In these methods, the obtained splines had grain size above 15 nm. Size distributions ranging from o15 to 100 nm were observed. Information was added in the manuscript

Line 115. So exactly 6 nm?  And is there really no size distribution?        

The grain size was in range of 6 to 7 nm.

Correction was made in the manuscript.

  1. How stable were the SEM and TEM pictures with time and were there any features indicating aging of the obtained materials?

SEM and TEM images were taken about a week after the synthesis. No features indicating that anything was happening to the material were noticed. However we did not perform any aging tests, nor visualizing the obtained materials with SEM and TEM later.

Round 2

Reviewer 1 Report

The conclusion of this paper is of certain significance for hydrothermal oxide synthesis. However, compared with the previous manuscript, unfortunately, the authors have not made much improvement. The figures and writing of this manuscript are still need to be improved to make readers have a better reading experience. Please take more time to improve the quality of the manuscript.

Author Response

REVIEWER 1 (ROUND 2)

The conclusion of this paper is of certain significance for hydrothermal oxide synthesis. However, compared with the previous manuscript, unfortunately, the authors have not made much improvement. The figures and writing of this manuscript are still need to be improved to make readers have a better reading experience. Please take more time to improve the quality of the manuscript.

The drawings have been corrected. Text has been checked in order to make it more readable and informative.

Reviewer 2 Report

The authors have significantly improved the manuscript according to the recommendations of the reviewers, so the manuscript can be recommended for publication.

Author Response

REVIEWER 2 (ROUND 2)

The authors have significantly improved the manuscript according to the recommendations of the reviewers, so the manuscript can be recommended for publication.

Thank you for the comment.
